# Mitogenomic Analysis and Phylogenetic Implications for the Deltocephaline Tribe Chiasmini (Hemiptera: Cicadellidae: Deltocephalinae)

**DOI:** 10.3390/insects15040253

**Published:** 2024-04-08

**Authors:** Bismillah Shah, Muhammad Asghar Hassan, Bingqing Xie, Kaiqi Wu, Hassan Naveed, Minhui Yan, Christopher H. Dietrich, Yani Duan

**Affiliations:** 1Anhui Province Key Laboratory of Integrated Pest Management on Crops, Key Laboratory of Biology and Sustainable Management of Plant Diseases and Pests of Anhui Higher Education Institutes, School of Plant Protection, Anhui Agricultural University, Hefei 230036, China; bismillahshah1990@yahoo.com (B.S.); xbq17692331701@163.com (B.X.); fyy6697@sina.com (K.W.); minhui2022@126.com (M.Y.); 2Department of Forestry Protection, School of Forestry and Biotechnology, Zhejiang A&F University, 666 Wusu Street, Linan, Hangzhou 311300, China; 3The Provincial Special Key Laboratory for Development and Utilization of Insect Resources, Institute of Entomology, Guizhou University, Guiyang 550025, China; kakojan112@gmail.com; 4School of Life Sciences, Jiangsu University, Zhenjiang 212013, China; hassan.naveed88@outlook.com; 5Illinois Natural History Survey, Prairie Research Institute, University of Illinois, Champaign, IL 61820, USA; chdietri@illinois.edu

**Keywords:** Hemiptera, Auchenorrhyncha, Cicadellidae, phylogeny, molecular biology

## Abstract

**Simple Summary:**

The mitochondrial genome of insects has been widely used in studies of molecular evolution, population genetics, phylogenetics, and species identification due to its small size (approximately 14–20 kb), relatively high evolutionary rate, low levels of recombination, and high genome copy numbers. In this study, 13 complete mitogenomes of Chiasmini, a diverse group of grassland leafhoppers, were sequenced and analyzed for the first time. The phylogenetic position of Chiasmini within leafhoppers and the phylogenetic relationships among Chiasmini genera were reconstructed. The results show that all 13 mitogenomes are composed of a circular, double-stranded molecule that consists of 37 genes with a total length ranging from 14,805 to 16,269 bp and a variable number of non-coding A + T-rich regions. The gene size, order, arrangement, base composition, codon usage, and secondary structure of tRNAs in these newly sequenced mitogenomes of 13 species are highly conserved in Chiasmini. Phylogenetic analysis of the newly sequenced genomes plus representatives of other tribes and subfamilies of Cicadellidae recover all included subfamilies of Cicadellidae and tribes of Deltocephalinae as monophyletic, except Athysanini and Opsiini in Deltocephalinae, which are paraphyletic in agreement with some other recent studies. Chiasmini form a monophyletic group consisting of seven monophyletic genera arranged as follows: ((*Zahniserius* + (*Gurawa* + (*Doratura* + *Aconurella*))) + (*Leofa* + (*Exitianus* + *Nephotettix*))).

**Abstract:**

The grassland leafhopper tribe Chiasmini (Cicadellidae: Deltocephalinae) presently comprises 324 described species worldwide, with the highest species diversity occurring in the Nearctic region but a greater diversity of genera occurring in the Old World. In China, this tribe comprises 39 described species in 11 genera, but the fauna remains understudied. The complete mitogenomes of three species of this tribe have been sequenced previously. In order to better understand the phylogenetic position of Chiasmini within the subfamily Deltocephalinae and to investigate relationships among Chiasmini genera and species, we sequenced and analyzed the complete mitogenomes of 13 species belonging to seven genera from China. Comparison of the newly sequenced mitogenomes reveals a closed circular double-stranded structure containing 37 genes with a total length of 14,805 to 16,269 bp and a variable number of non-coding A + T-rich regions. The gene size, gene order, gene arrangement, base composition, codon usage, and secondary structure of tRNAs of the newly sequenced mitogenomes of these 13 species are highly conserved in Chiasmini. The ATN codon is commonly used as the start codon in protein-coding genes (PCGs), except for *ND5* in *Doratura* sp. and *ATP6* in *Nephotettix nigropictus*, which use the rare GTG start codon. Most protein-coding genes have TAA or TAG as the stop codon, but some genes have an incomplete T stop codon. Except for the tRNA for serine (*trnS1*(AGN)), the secondary structure of the other 21 tRNAs is a typical cloverleaf structure. In addition to the primary type of G–U mismatch, five other types of tRNA mismatches were observed: A–A, A–C, A–G, U–C, and U–U. Chiasmini mitochondrial genomes exhibit gene overlaps with three relatively stable regions: the overlapping sequence between *trnW* and *trnC* is AAGTCTTA, the overlapping sequence between *ATP8* and *ATP6* is generally ATGATTA, and the overlapping sequence between *ND4* and *ND4L* is generally TTATCAT. The largest non-coding region is the control region, which exhibits significant length and compositional variation among species. Some Chiasmini have tandem repeat structures within their control regions. Unlike some other deltocephaline leafhoppers, the sequenced Chiasmini lack mitochondrial gene rearrangements. Phylogenetic analyses of different combinations of protein-coding and ribosomal genes using maximum likelihood and Bayesian methods under different models, using either amino acid or nucleotide sequences, are generally consistent and also agree with results of prior analyses of nuclear and partial mitochondrial gene sequence data, indicating that complete mitochondrial genomes are phylogenetically informative at different levels of divergence within Chiasmini and among leafhoppers in general. Apart from Athysanini and Opsiini, most of the deltocephaline tribes are recovered as monophyletic. The results of ML and BI analyses show that Chiasmini is a monophyletic group with seven monophyletic genera arranged as follows: ((*Zahniserius* + (*Gurawa* + (*Doratura* + *Aconurella*))) + (*Leofa* + (*Exitianus* + *Nephotettix*))).

## 1. Introduction

Insect mitochondrial genomes are typically small double-stranded circular molecules containing 37 genes, including 13 protein-coding genes (PCGs), 22 transfer RNAs (tRNAs) genes, 2 ribosomal RNAs (rRNAs) genes, and a control region (CR) or non-coding A + T-rich region [1]. The mitogenome is a suitable molecular marker for molecular evolution, phylogenetic relationships, and species identification due to its smaller size (approximately 14–20 kb), maternal inheritance, relatively high evolutionary rate, low levels of recombination, high genome copy numbers, and conservative gene components [2,3,4].

The grassland-associated leafhopper tribe Chiasmini (Cicadellidae: Deltocephalinae) presently comprises 324 described species in 21 genera distributed worldwide, with the greatest reported species diversity occurring in the Nearctic but higher genus-level diversity in the Oriental Region [5,6,7]. In China, Chiasmini comprises 39 described species in 11 genera: *Aconurella* Ribaut (11 species), *Nephotettix* Matsumura (6 spp.), *Doratura* Sahlberg (4 spp.), *Leofa* Distant (3 spp.), *Gurawa* Distant (4 spp.), *Doraturopsis* Lindberg (3 spp.), *Exitianus* Ball (3 spp.), *Baileyus* Singh-Pruthi (2 spp.), *Chiasmus* Mulsant & Rey (2 spp.), and *Zahniserius* Duan & Zhang (1 sp.) [5]. However, 13 species records are doubtful due to a lack of specimens [5,8,9,10,11,12,13,14,15]. Many species of the subfamily Deltocephalinae, including Chiasmini, are important agricultural pests, causing damage by piercing and sucking plant sap and transmitting plant pathogens [6]. Most members of the Chiasmini tribe feed on perennial grasses or sedges in native grasslands, but *Nephotettix* species are known to cause damage to cultivated rice, wheat, and barley, causing weakened growth, slowed development, and even the death of entire plants [15]. *Exitianus* species also transmit various pathogens affecting crops and forage grasses, such as maize dwarf mosaic disease and Bermuda grass white leaf disease [16,17].

Since the establishment of this tribe, various taxonomic changes have been proposed for the adjustment of various genera and species based on external body morphology, male and female genitalia, and molecular data [5,18,19,20,21,22,23]. The first phylogeny of Deltocephalinae, based on adult morphological characters and molecular data (two nuclear genes, 28S and H3), included only seven representatives of Chiasmini and did not recover the tribe as monophyletic. A subsequent more-detailed phylogenetic study of this tribe, including 21 genera and 316 species, incorporating molecular data from four genes (two mitochondrial, COII and ND1, and two nuclear genes, 28S and H3) also failed to consistently recover the tribe as monophyletic [23]. However, the recent comprehensive phylogenomic analysis of Deltocephalinae, Cao et al. [7] comprising 730 terminal taxa and >160,000 nucleotide positions obtained through anchored hybrid enrichment strongly supported the monophyly of Chiasmini and grouped New and Old World taxa into distinct lineages. Stenometopiini was recovered as the sister group of Chiasmini in the coalescent gene tree analysis, but concatenated maximum likelihood analyses suggested that the sister group of Chiasmini includes Stenometopiini plus three other grass-specialist tribes. In the most recent study, Zhang et al. [5] reconstructed the phylogenetic relationships of 20 Chinese chiasmine species from eight genera using four genes: two mitochondrial (COI and 16S) and two from the nuclear genome (28S and H3). Their results consistently supported the monophyly of Chiasmini and recovered stable relationships between the genera and species of Chiasmini, but some relationships were incongruent with those obtained in previous analyses. The phylogeny of this tribe has not yet been analyzed using full mitochondrial genome sequences. Mitochondrial genome sequences have been shown to be highly informative of relationships among other groups of deltocephaline leafhoppers, and some deltocephalines have mitochondrial gene rearrangements that provide additional phylogenetically informative characters [24]. Unfortunately, only three complete mitogenomes of Chiasmini have been sequenced and analyzed so far, so the value of such data for elucidating relationships within this tribe remains unexplored [25,26,27]. The taxonomy of the Chinese Chiasmini is well-studied [8,9,10,11,12,13,14,15], but complete mitogenome data are scarcely available. This study aimed to provide 13 new mitogenome sequences of the tribe Chiasmini, conduct comparative mitogenomic analyses, reconstruct the phylogenetic relationships between Chiasmini and other tribes in the subfamily Deltocephalinae, and estimate within the tribe using the data from published sources and newly sequenced mitogenomes.

## 2. Materials and Methods

### 2.1. Taxon Sampling

Leafhopper samples included in this study were collected between 2010 and 2019 from various collection sites in Anhui, Guangxi, Yunnan Provinces, and the Xinjiang Autonomous Region, China (Table 1). Fresh samples were initially preserved in 95% or 100% ethanol and subsequently stored at −80 °C before the study. Prior to DNA extraction, the samples were identified at the species level based on external morphology and male genitalia using the available taxonomic literature [8,9,10,11,12,13,14,15]. The external body morphology and male genitalia of each species were examined using Nikon SMZ1500 (Nikon Corporation, Tokyo, Japan) and Motic K-700HS microscopes (MacAudi Electric Co., Ltd., Xiamen, China). All specimens studied are listed in Table 1 and are deposited at Northwest A&F University, Yangling, China (NWAFU), and Anhui Agricultural University, Hefei, China (AAU).

### 2.2. DNA Extraction and Sequencing

Genomic DNA was extracted from ethanol-preserved specimens following the standard protocol of the Qiagen DNeasy Blood and Tissue kit. The concentration and quality of DNA were determined by a Nanodrop 2000 Spectrophotometer (Thermo Scientific, Waltham, MA, USA) and 1% agarose gel electrophoresis. The mitogenomes were sequenced using high-throughput sequencing on an Illumina NovaSeq platform (United States Illumina Company, San Diego, CA, USA) for each species, generating approximately 2 GB of data per sample with a read length of 150 bp. The raw data were subjected to quality control and filtering (removing adapter contamination, trimming reads shorter than 50 bp, removing reads with an average quality score lower than 20, and removing reads with more than 3 Ns), resulting in high-quality data for subsequent analysis.

### 2.3. Mitogenome Sequence Assembly and Annotation

The mitogenome sequences were assembled using NOVOPlasty software under the Linux system [28]. The assembled sequences were then aligned in GENEIOUS software v. 10.2.3 (https://www.geneious.com/, accessed on 12 December 2023) [29] to process degenerate bases and detect circularity, resulting in a complete mitogenome sequence.

The assembled sequences were compared with published mitochondrial genome sequences of other chiasmine leafhopper species (*Aconurella prolixa* [MZ433366], *Exitianus indicus* [KY039128], and *Nephotettix cincticeps* [KP749836]) using the built-in MAFFT plugin in GENEIOUS. The sequences of 37 individual genes, including the 13 PCGs, 22 tRNAs, and 2 rRNAs, were extracted from each mitogenome. The lengths and positions of the 13 protein-coding genes were confirmed by comparing them to multiple reference sequences, and the start and stop codons were identified within the open reading frames to annotate the protein-coding genes. The annotation of RNA genes and the prediction of tRNA secondary structure were performed using the Mitos online tool (http://mitos.bioinf.uni-leipzig.de/index.py, accessed on 12 December 2023) [30]. The assembled mitochondrial genome sequence was uploaded and analyzed using the invertebrate mitochondrial codon table to determine RNA genes’ start and stop positions. The prediction of tRNA secondary structures was also conducted on the same website.

### 2.4. Mitogenome Sequence Analysis

The relative synonymous codon usage (RSCU) for PCGs and nucleotide composition were calculated using PhyloSuite software (version 1.2.2). The annotated sequence file was imported into PhyloSuite to extract annotations. Tandem Repeats Finder software was used to predict tandem repeats in the control region under default settings [31]. The genetic distances of the protein-coding gene sequences among samples were analyzed using MEGA software (version 7.0) with the Kimura 2-parameter model. Non-synonymous and synonymous substitution rates and sliding window analyses of protein-coding gene sequences were calculated using DnaSP software (version 3.12.03) [32] using default settings, with a 200 bp sliding window and 20 bp overlap.

After removing stop codons, protein-coding genes, and rRNA genes were aligned using MAFFT software (version 7.520) [33,34] with G-strategy and Q-strategy for protein-coding genes and rRNA genes, respectively. Gblocks software (version 0.91b) was used to remove gaps and ambiguous positions under default settings [35]. Concatenation of the aligned sequence files was performed using PhyloSuite software (version 1.2.2). Substitution saturation was calculated using DAMBE software (version 6.0.0) [36] by importing the concatenated sequence files and selecting the Xia method. The heterogeneity of different combination datasets was analyzed using AliGROOVE software (version 1.0.8) [37], with different sequence files being imported and selecting Ambiguity for nucleotide sequences and BLOSUM62 for amino acid sequences.

### 2.5. Phylogenetic Analysis

PartitionFinder software 2.1.1 [38] was used to find the best partitioning scheme and the best-fitting substitution models for the first, second, and third codon positions of protein-coding genes and amino acid sequences (Appendix A). The maximum likelihood (ML) tree was constructed using IQ-TREE software (version 2.2.0) [39] by importing the partitioned model, selecting the ultrafast bootstrap (UFB) algorithm, and evaluating the support of each node based on 1000 replicates. The Bayesian (BI) tree was estimated using MrBayes software (version 3.2.2) [40] by importing the partitioned model and running 4 MCMC chains for 40 million generations, with a sampling frequency of 1000 generations. After convergence, as determined by an average standard deviation below 0.01, the first 25% of trees were discarded, and a consensus tree was calculated to evaluate the posterior probability of each node.

## 3. Results and Discussion

### 3.1. Mitogenome Analysis

In this study, a total of 13 new mitogenomes were sequenced for leafhoppers of the tribe Chiasmini from China. Comparative mitochondrial genomic analysis was performed based on both newly sequenced and three previously published mitogenomes, resulting from in a total of 16 Chiasmini species included in the subsequent phylogenetic analysis (Table 1).

### 3.2. Mitogenome Organization and Gene Content

All 16 Chiasmini mitochondrial genomes contain 37 genes, including 13 protein-coding genes, 22 transfer RNAs, 2 ribosomal RNAs, and a relatively long control region. The total length of the 16 Chiasmini mitochondrial genomes ranges from 14,805 to 16,269 bp, with the longest being *Doratura homophyla* (16,269 bp) and the shortest being *Nephotettix cincticeps* (14,805 bp) (Table 2). The 13 protein-coding genes range in combined length from 10,869 to 10,980 bp. The 22 tRNA genes range in combined length from 1433 to 1475 bp. The *rrnL* gene ranges in length from 1193 to 1216 bp. The *rrnS* gene ranges in length from 737 to 801 bp. The control region (CR) is by far the most variable, ranging in length from 399 to 1828 bp, and is the main factor affecting variation in mitochondrial genome size. Each newly sequenced mitogenome contains a typical set of 37 mitochondrial genes, 13 PCGs, 22 tRNAs, 2 rRNAs, and 1 control region (Figure 1 and Figure 2). We did not detect any gene rearrangements within Chiasmini; mitochondrial gene rearrangements are apparently rare among leafhoppers but have been reported in some other tribes of Deltocephalinae [24].

### 3.3. Nucleotide Composition

The AT content of the 16 Chiasmini mitochondrial genomes ranges from 73.6% to 77.4%, and each component exhibits a relatively high AT content. Protein-coding genes (PCGs) have the lowest AT content, followed by tRNAs, *rrnS*, and *rrnL*. The highest AT content is found in the control region (CR), with a mean value exceeding 81%. This indicates a clear AT bias in the complete mitogenome in Chiasmini species (Table 2 and Figure 3).

### 3.4. Protein-Coding Genes and Codon Usage

The protein-coding genes (PCGs) of the Chiasmini species have a total of six initiation codons: ATG (88 times), ATA (47 times), ATT (45 times), TTG (16 times), ATC (9 times), and GTG (3 times). Regarding the stop codons, the majority are TAA or TAG, but some are T. Specifically, TAA is used 167 times, TAG is used 24 times, and T is used 17 times (Table 3). Statistical results of the relative synonymous codon usage (RSCU) show that the frequently used amino acids in the Chiasmini mitochondrial genome are Ile, Leu2, Phe, and Met (Figure 4).

Based on the sliding window analysis of 13 PCGs in the Chiasmini 16 species, the nucleotide diversity of *ATP8* (Pi = 0.345) is the highest. At the same time, *CYTB* (Pi = 0.199), *ND1* (Pi = 0.198), *COX3* (Pi = 0.197), and *COX1* (Pi = 0.169) show lower nucleotide diversity (Figure 5). According to a genetic distance analysis of the 13 PCGs, the *ATP8* (0.45) gene has the highest genetic distance and fastest evolution rate, and the *COX1* (0.19) gene has the lowest genetic distance and slowest evolution rate. The ratio of synonymous (Ks) to non-synonymous (Ka) substitution rates (Ka/Ks, ω) among the 13 PCGs ranges from 0.09 to 0.696 (0 < ω < 1). All ω values are less than 1, indicating that the 13 PCGs are under purifying selection at the gene level. *COX1* has the lowest ω value (0.09), while *ATP8* has the highest ω value (0.70), showing a significant difference (Figure 6).

### 3.5. RNA Genes

All 22 tRNA genes of the Chiasmini mitochondrial genome range in length from 60 to 70 bp. The dihydrouridine (DHU) arm of *trnS1* (AGN) is missing and forms a circular structure (Figure 7), while the remaining 21 tRNAs are predicted to have a typical cloverleaf structure (Appendix A). In addition to A–U and G–C pairing, the predicted secondary structure of tRNA includes five mismatched types: A–A, A–C, A–G, U–C, and U–U (Table 4). The *rrnL* length ranges from 1193 to 1216 bp, and the *rrnS* length ranges from 737 to 801 bp in the rRNA (Table 4).

### 3.6. Gene Overlap and Non-Coding Regions

The gene arrangement of the Chiasmini mitochondrial genome is relatively compact, with overlapping regions ranging from 6 bp (in *Nephotettix nigropictus*) to 12 bp (in *Leofa pulchella*). There are three stable gene overlaps: the overlapping sequence of *trnW* and *trnC* is AAGTCTTA, the overlapping sequence of *ATP8* and *ATP6* is generally ATGATTA, and the overlapping sequence of *ND4* and *ND4L* is generally TTATCAT. The Chiasmini mitochondrial genome also contains intergenic spacer regions, ranging from 11 bp (in *Aconurella diplachnis*) to 19 bp (in *Nephotettix nigropictus*). Except for a 548 bp-long spacer between *trnM* and *ND2* in *Exitianus indicus*, the spacer lengths are generally small, usually less than 30 bp.

### 3.7. Phylogenetic Relationships

This study aimed to reconstruct the phylogenetic relationships among the cicadellid subfamilies, tribes in the subfamily Deltocephalinae, and the genera and species of Chiasmini. Therefore, we selected all available Deltocephalinae species and no more than two representative species from each other subfamily instead of the entire 153 cicadellid mitogenomes currently available from NCBI. In this phylogenetic analysis, we selected 65 Membracoidea species (including 62 leafhoppers and 3 treehoppers) as the ingroup and 2 Cicadoidea and 2 Cercopoidea species as the outgroups (Table 5).

Mitogenome data of the 65 species representing 14 subfamilies of Cicadellidae were concatenated into five datasets: 13 PCG sequences (P123); 13 PCG and 2 rRNA sequences (P123R); 13 PCG sequences with the third codon position removed (P12); 13 PCG sequences with the third codon position removed, as well as 2 rRNA sequences (P12R); and amino acid sequences (AA). Maximum likelihood (ML) and Bayesian inference (BI) methods were used for phylogenetic analysis, and tests of base substitution saturation indicated that the selected datasets were suitable for phylogenetic analysis. There were no significant differences in sequence divergence among the different datasets, and removing the third codon position in the nucleotide datasets reduced heterogeneity between species. Based on 10 phylogenetic trees reconstructed herein using five datasets and two analysis methods (Figure 8 and Figure 9; Appendix A), the included representatives of each subfamily for which more than one representative was included (Coelidiinae, Deltocephalinae, Eurymelinae, Evacanthinae, Hylicinae, Iassinae, Ledrinae, Megophthalminae, Mileewinae, and Typhlocybinae) form monophyletic groups (bootstrap values, BS = 100; posterior probabilities, PP = 1.0). As in other recent molecular phylogenetic analyses, branches pertaining to relationships among leafhopper subfamilies received only low to moderate support and were unstable among analyses, with the exception of Megophthalminae consistently grouping with the treehopper lineage (Membracidae + Aetalionidae) with maximum support. Ledrinae is the earliest diverging cicadellid subfamily in both AA-ML and AA-BI trees (Figure 8), while in the remaining eight phylogenetic trees, Typhlocybinae is sister to the remaining leafhoppers (Figure 9; Appendix A). In the P12R-BI, AA-ML, and AA-BI trees, (Hylicinae + (Iassinae + Coelidiinae)) formed a clade, sister to Deltocephalinae, but the support for this node was generally low (P12R-BI, PP = 0.39; AA-ML, BS = 62; AA-BI, PP = 0.71). In the remaining seven phylogenetic trees, Hylicinae formed a clade sister to Deltocephalinae, with support in P123-BI and P123R-BI analyses of 0.93 and 0.86 (PP), respectively.

Within the Deltocephalinae, the phylogenies based on mitogenome sequence data are consistent with the recent anchored-hybrid-based phylogeny of Cao et al. [7], except in a few parts of the tree that received less than maximum branch support. As in the Cao et al. [7] analysis, Fieberiellini, Penthimiini, and *Tambocerus* represent early diverging lineages, and the tribe Athysanini is polyphyletic. *Orosius orientalis* of tribe Opsiini is separate from other opsiines (*Hishimonoides recurvatis* + *Japananus hyalinus*), indicating the non-monophyly of Opsiini, but the branches separating the included opsiines received low support (0.35–0.45) in our Bayesian analysis of amino acid sequences.

The seven included genera within Chiasmini consistently form a strongly supported clade, with the phylogenetic relationships supporting a stable sister-group relationship between *Exitianus* and *Nephotettix*, consistent with their morphological similarity, with *Leofa* being the closest relative. The four species of *Aconurella*, which are difficult to distinguish based on morphological characters, are also grouped into a larger clade, which shows a close sister-group relationship with *Zahniserius*, *Doratura*, and *Gurawa*, presenting a topology of ((*Zahniserius* + (*Gurawa* + (*Doratura* + *Aconurella*))) + (*Leofa* + (*Exitianus* + *Nephotettix*))). The phylogenetic results obtained from the two different analytical methods do not vary significantly, and relationships within this clade are well-resolved under a homogenous substitution model; therefore, we chose to perform subsequent phylogenetic analysis under the homogenous model only. All analyses yielded results largely consistent with previous phylogenetic studies of this tribe by Zahniser and Dietrich [23], Gao et al. [15], Cao et al. [7], and Zhang et al. [5].

The 10 phylogenetic results based on the five datasets and two analysis methods (Figure 8 and Figure 9; Appendix A) indicate that mitochondrial genome sequences are informative of relationships within and among tribes of deltocephaline leafhoppers and within the tribe Chiasmini (bootstrap values, BS = 100; posterior probabilities, PP = 1.0). Within Chiasmini, relationships among the seven included genera (*Aconurella*, *Doratura*, *Exitianus*, *Gurawa*, *Leofa*, *Nephotettix*, and *Zahniserius*) are consistent among the 10 different topologies, with high support values (BS = 100, PP = 1.0) with one exception. In the PCG123-ML/BI and PCG123R-ML/BI trees, the topology is ((*Zahniserius* + *Gurawa*) + (*Doratura* + *Aconurella*)) + (*Leofa* + (*Exitianus* + *Nephotettix*)), while in the PCG12-ML/BI, PCG12R-ML/BI, and AA-ML/BI trees, the topology is ((*Zahniserius* + (*Gurawa* + (*Doratura* + *Aconurella*))) + (*Leofa* + (*Exitianus* + *Nephotettix*))). The latter topology has much higher support than the former and is more consistent with previous analyses based on combined mitochondrial and nuclear gene sequences (Zahniser and Dietrich [23]) and anchored hybrid nuclear gene data (Cao et al. [7]), suggesting that the phylogenetic relationships within Chiasmini based on the mitochondrial P12, PCG12R, and AA datasets are more reliable.

## 4. Conclusions

Comparative analysis of the complete mitogenomes of 16 species representing seven genera of the grass-specialist leafhopper tribe Chiasmini indicates a highly conserved overall genome structure and composition in this group, without any of the gene rearrangements reported for some other groups of deltocephaline leafhoppers. Overall, our phylogenetic results suggest that analyses of mitogenome sequence data provide good resolution of both deep and shallow branches within the phylogeny of Chiasmini and leafhoppers in general and yield results similar to analyses based on nuclear genes alone or combined nuclear and mitochondrial genes. Despite these results, the mitogenome sequences currently available represent only a tiny fraction of known species. More sequencing efforts are needed to increase the size of the available taxon sample and facilitate further exploration of the phylogeny of leafhoppers.

## Figures and Tables

**Figure 1 insects-15-00253-f001:**
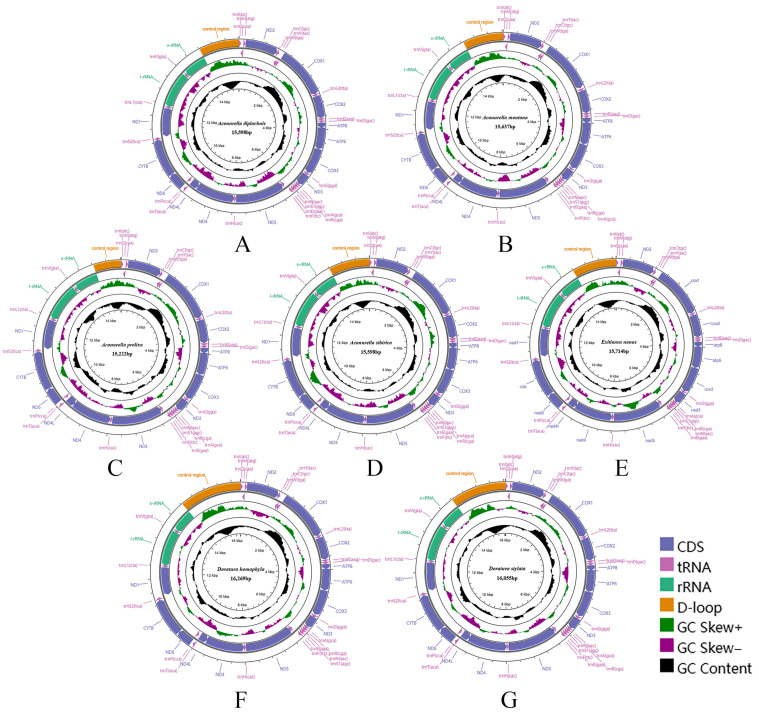
Circular maps of the complete mitochondrial genomes of Chiasmini species. (**A**) *Aconurella diplachnis*; (**B**) *Aconurella montana*; (**C**) *Aconurella prolixa*; (**D**) *Aconurella sibirica*; (**E**) *Exitianus nanus*; (**F**) *Doratura homophyla*; (**G**) *Doratura stylata*.

**Figure 2 insects-15-00253-f002:**
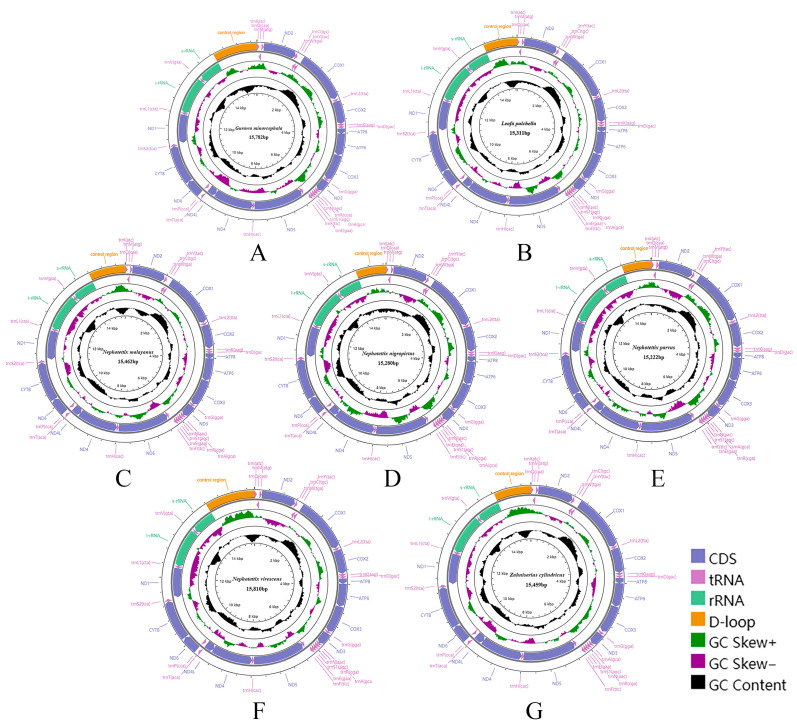
Circular maps of the complete mitochondrial genomes of Chiasmini species. (**A**) *Gurawa minorcephala*; (**B**) *Leofa pulchella*; (**C**) *Nephotettix malayanus*; (**D**) *Nephotettix nigropictus*; (**E**) *Nephotettix parvus*; (**F**) *Nephotettix virescens*; (**G**) *Zahniserius cylindricus*.

**Figure 3 insects-15-00253-f003:**
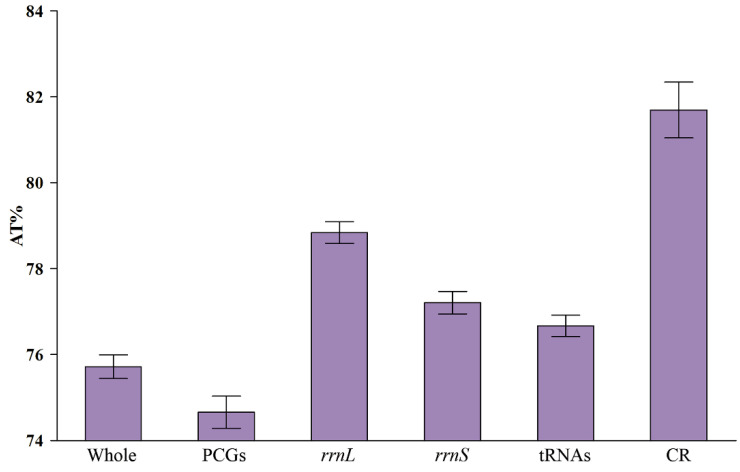
AT% in different regions of Chiasmini mitochondrial genomes.

**Figure 4 insects-15-00253-f004:**
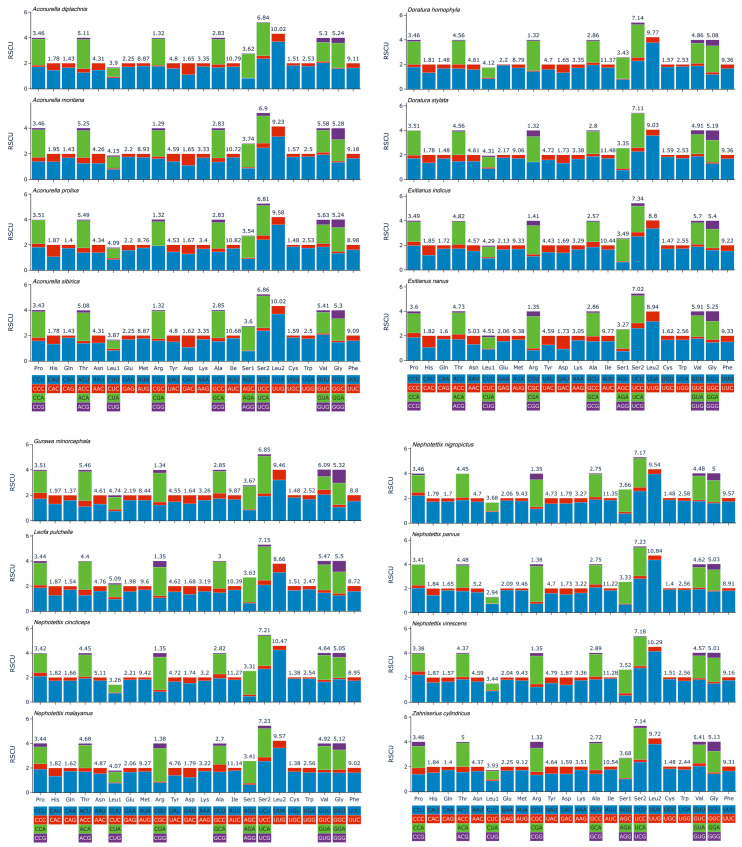
Relative synonymous codon usage (RSCU) of the PCGs of species of Chiasmini.

**Figure 5 insects-15-00253-f005:**
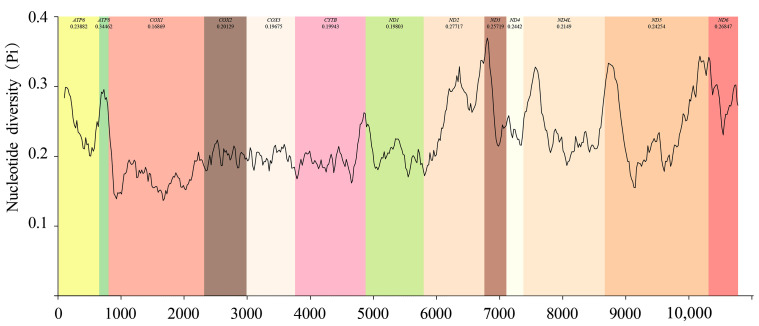
Sliding-window analyses based on 13 aligned PCGs among species of Chiasmini.

**Figure 6 insects-15-00253-f006:**
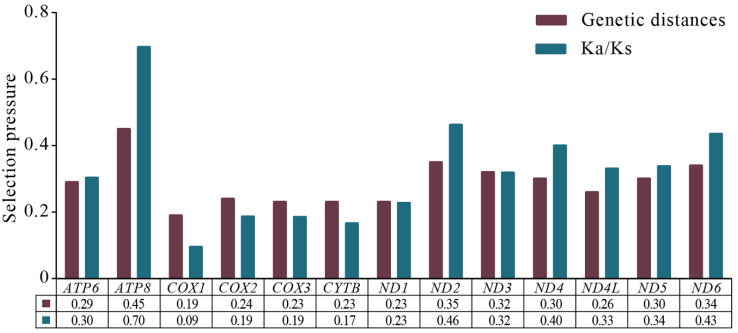
Genetic distances and ratios of non-synonymous (Ka) to synonymous (Ks) substitution rates of 13 aligned PCGs among species of Chiasmini. The average value for each PCG is shown below the gene name.

**Figure 7 insects-15-00253-f007:**
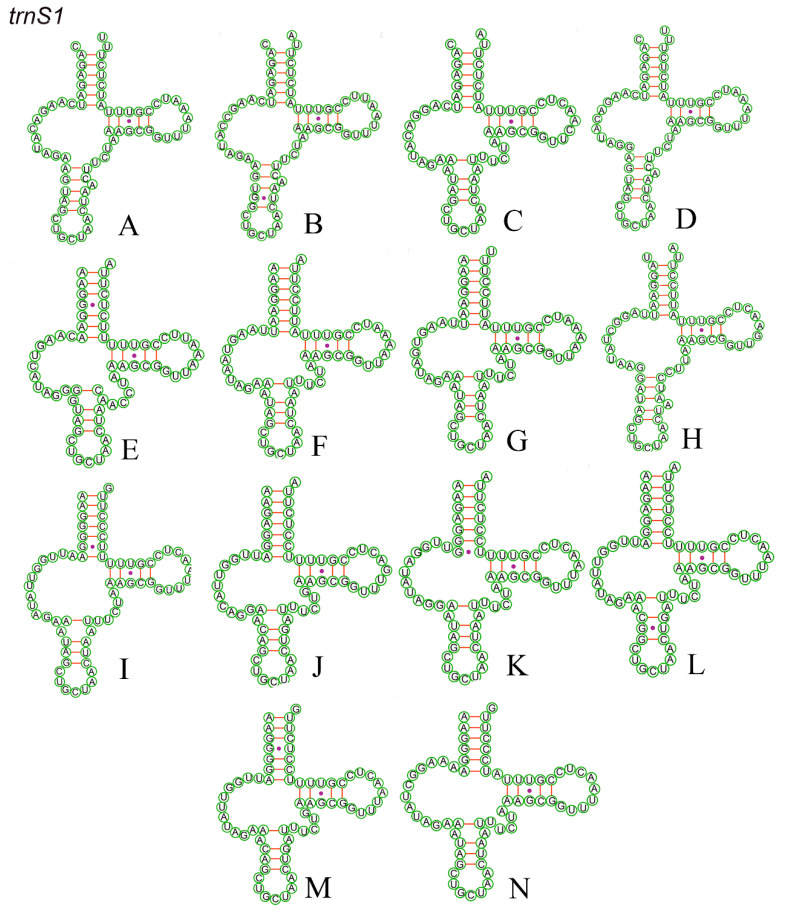
Putative secondary structures of the *trnS1* in 14 Chiasmini mitogenomes that do not possess a DHU-arm. Dashes (−) indicate Watson−Crick base pairing and dots (•) indicate G−U pairing. (**A**) *Aconurella diplachnis*; (**B**) *Aconurella montana*; (**C**) *Aconurella prolixa*; (**D**) *Aconurella sibirica*; (**E**) *Exitianus nanus*; (**F**) *Doratura stylata*; (**G**) *Doratura homophyla*; (**H**) *Gurawa minorcephala*; (**I**) *Leofa pulchella*; (**J**) *Nephotettix malayanus*; (**K**) *Nephotettix nigropictus*; (**L**) *Nephotettix parvus*; (**M**) *Nephotettix virescens*; (**N**) *Zahniserius cylindricus*.

**Figure 8 insects-15-00253-f008:**
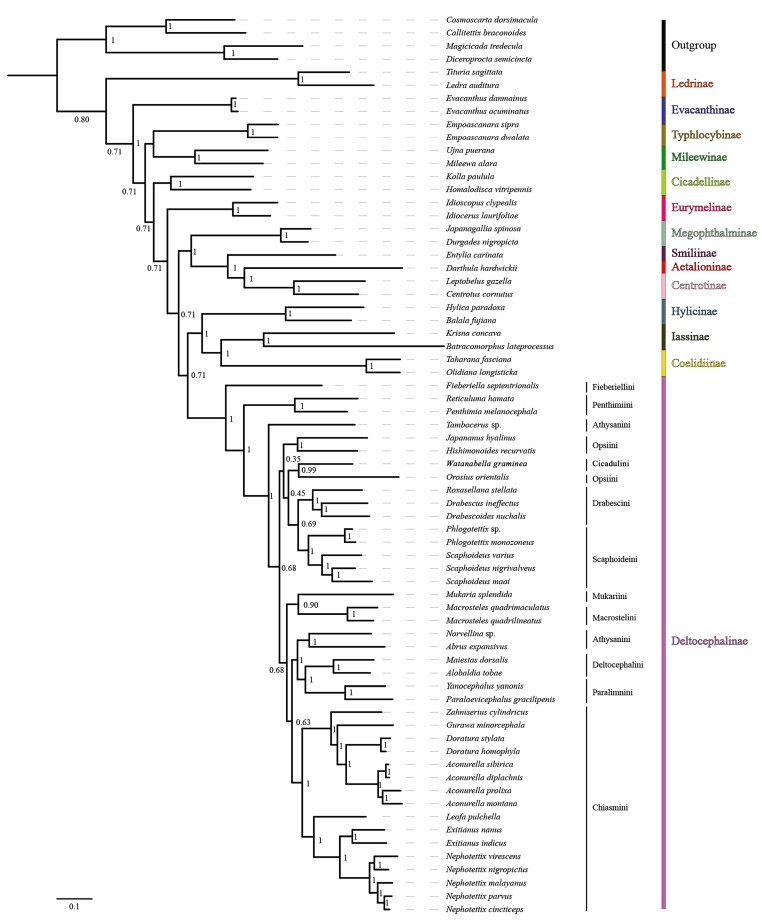
The phylogenetic relationships using the Bayesian inference (BI) analysis method based on the AA datasets. Numbers on each node correspond to the posterior probability (PP) values.

**Figure 9 insects-15-00253-f009:**
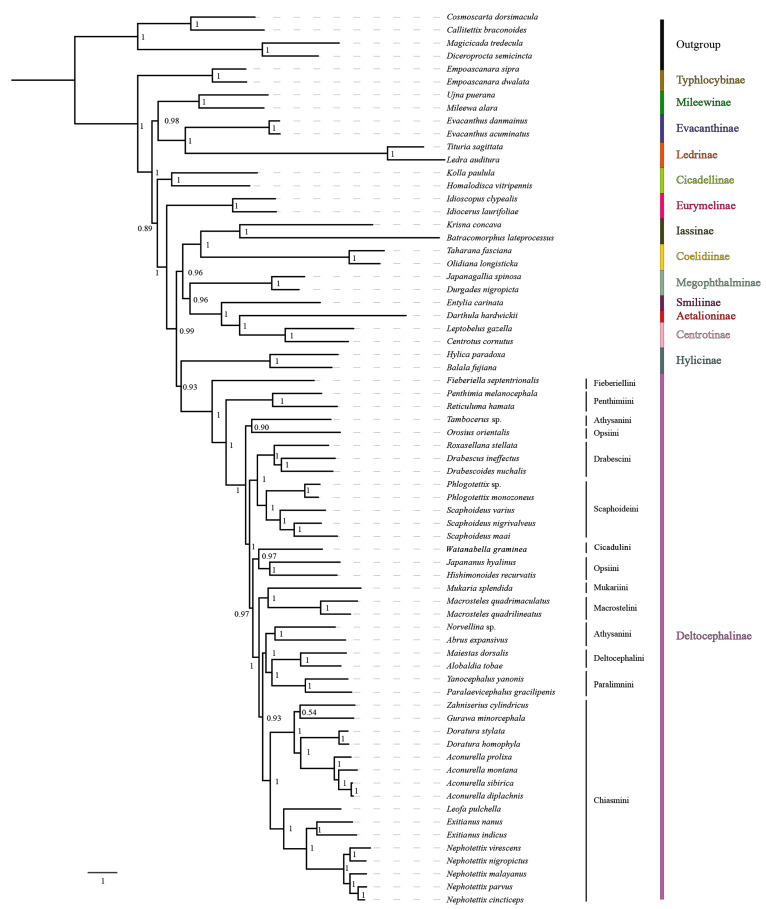
The phylogenetic relationships using the Bayesian inference (BI) analysis method based on the PCG datasets. Numbers on each node correspond to the posterior probability (PP) values.

**Table 1 insects-15-00253-t001:** Specimens and previously published data used in this study (all from China).

Organism	Locality	Code	GenBank Accession	Reference
*Aconurella diplachnis*	Xinjiang	Hm086374	OK105069	This study
*Aconurella montana*	Yunnan	Hm087561	OK105070	This study
*Aconurella prolixa*	Guangxi	HN027	MZ433366	[26]
*Aconurella sibirica*	Anhui	Hm083232	OK105071	This study
*Doratura homophyla*	Xinjiang	HN010	OK105076	This study
*Doratura stylata*	Xinjiang	HN029	OK105077	This study
*Exitianus indicus*	Henan		KY039128	[25]
*Exitianus nanus*	Guangxi	YN002	OK105078	This study
*Gurawa minorcephala*	Guangxi	Hm086937	OK105072	This study
*Leofa pulchella*	Guangxi	GX031	OK105073	This study
*Nephotettix cincticeps*	Henan		KP749836	[25]
*Nephotettix malayanus*	Guangxi	GX073	OK105079	This study
*Nephotettix nigropictus*	Yunnan	YN041	OK105080	This study
*Nephotettix parvus*	Yunnan	Hm086799	OK105074	This study
*Nephotettix virescens*	Yunnan	YN–009	OK105081	This study
*Zahniserius cylindricus*	Yunnan	Hm080184	OK105075	This study

**Table 2 insects-15-00253-t002:** Length and base compositions of Chiasmini mitochondrial genomes.

Organism	Length (bp)	AT%	AT Skew	GC Skew
*Aconurella diplachnis*	15,598	76.4	0.123	−0.186
*Aconurella montana*	15,637	75.2	0.128	−0.169
*Aconurella prolixa*	15,222	75.2	0.125	−0.169
*Aconurella sibirica*	15,598	76.0	0.124	−0.183
*Doratura homophyla*	16,269	76.9	0.090	−0.160
*Doratura stylata*	16,055	76.3	0.093	−0.165
*Exitianus indicus*	16,089	75.1	0.119	−0.157
*Exitianus nanus*	15,714	74.3	0.133	−0.144
*Gurawa minorcephala*	15,782	73.6	0.128	−0.220
*Leofa pulchella*	15,311	74.6	0.129	−0.150
*Nephotettix cincticeps*	14,805	75.1	0.071	−0.094
*Nephotettix malayanus*	15,462	75.4	0.074	−0.089
*Nephotettix nigropictus*	15,280	77.0	0.065	−0.087
*Nephotettix parvus*	15,222	77.4	0.070	−0.106
*Nephotettix virescens*	15,810	77.3	0.069	−0.084
*Zahniserius cylindricus*	15,459	75.7	0.096	−0.136

**Table 3 insects-15-00253-t003:** The use of start and stop codons of mitochondrial PCGs in Chiasmini.

Species	Start Codon/Stop Codon (ATN/TAN)												
	*ATP6*	*ATP8*	*COX1*	*COX2*	*COX3*	*CYTB*	*ND1*	*ND2*	*ND3*	*ND4*	*ND4L*	*ND5*	*ND6*
*Aconurella diplachnis*	G/A	A/A	G/G	A/T-	G/A	G/A	T/A	A/A	A/A	G/T-	T/A	TTG/A	T/A
*Aconurella montana*	G/A	C/A	G/A	A/T-	G/A	G/G	T/A	A/A	A/A	G/T-	G/A	TTG/A	A/A
*Aconurella prolixa*	G/A	C/A	G/A	A/T-	G/A	G/G	C/A	A/A	A/A	G/T-	T/A	TTG/G	T/A
*Aconurella sibirica*	G/A	A/A	G/G	A/T-	G/A	G/A	T/A	A/A	A/A	G/T-	T/A	TTG/A	T/A
*Doratura homophyla*	G/A	A/A	G/A	A/A	G/A	G/A	T/A	A/A	T/A	G/T-	C/A	GTG/G	A/A
*Doratura stylata*	G/A	T/A	G/A	A/A	G/A	G/A	T/A	A/A	T/A	G/T-	T/A	GTG/G	A/A
*Exitianus indicus*	G/A	C/A	G/A	A/T-	G/A	G/A	T/A	T/A	A/A	A/T-	T/G	A/A	T/A
*Exitianus nanus*	G/A	C/A	G/G	A/A	G/A	G/A	T/A	T/A	A/G	G/T-	T/G	G/G	TTG/T-
*Gurawa minorcephala*	G/A	A/A	G/G	G/A	G/A	G/A	G/A	T/A	T/A	G/A	T/A	TTG/A	T/T-
*Leofa pulchella*	G/A	T/A	G/A	A/A	G/G	G/G	A/A	T/A	T/G	G/A	T/A	TTG/A	A/A
*Nephotettix cincticeps*	G/A	G/A	G/A	A/A	G/A	G/A	T/A	T/A	A/TA	G/A	G/G	A/A	A/A
*Nephotettix malayanus*	G/A	T/A	G/A	A/A	G/A	G/A	T/A	T/A	A/A	G/A	C/A	TTG/A	TTG/A
*Nephotettix nigropictus*	GTG/A	A/A	G/A	A/A	G/A	G/A	T/A	T/A	A/A	G/A	G/A	TTG/A	TTG/A
*Nephotettix parvus*	G/A	C/A	G/A	A/A	G/G	G/A	T/A	T/A	A/G	G/G	G/A	TTG/A	TTG/A
*Nephotettix virescens*	G/A	A/A	G/A	A/A	G/A	G/A	T/A	T/A	A/A	G/A	G/A	TTG/A	TTG/A
*Zahniserius cylindricus*	G/A	T/A	G/G	A/G	G/G	G/A	G/A	T/A	C/A	G/T-	T/A	TTG/A	T/T-

**Table 4 insects-15-00253-t004:** Comparison of transfer RNA and mitochondrial RNA in Chiasmini mitochondrial genomes.

Species	tRNA Base Mismatch						*rrnL* (bp)	*rrnS* (bp)
	A–A	A–C	A–G	G–U	U–C	U–U		
*Aconurella diplachnis*				25		2	1201	801
*Aconurella montana*	2	2		25		3	1203	783
*Aconurella prolixa*	1			27		1	1210	793
*Aconurella sibirica*	1			25		3	1206	780
*Doratura homophyla*	3			26	1	3	1205	743
*Doratura stylata*	2			21		5	1206	746
*Exitianus indicus*				29		5	1208	751
*Exitianus nanus*	1	1	1	21	1	5	1209	744
*Gurawa minorcephala*	1	1	1	31		3	1211	750
*Leofa pulchella*	2			25		3	1198	737
*Nephotettix cincticeps*	2		1	20		6	1201	741
*Nephotettix malayanus*	2			23		4	1193	745
*Nephotettix nigropictus*	2			18		4	1207	748
*Nephotettix parvus*	3			23		5	1206	745
*Nephotettix virescens*	2			23		3	1206	745
*Zahniserius cylindricus*	2			33		3	1216	741

**Table 5 insects-15-00253-t005:** Summary of mitochondrial genome information used in this study.

Superfamily/Family/ Subfamily	Tribe	Species	Accession Number	Reference
**Cercopoidea**				
Cercopidae				
Callitettixinae	Callitettixini	*Callitettix braconoides*	JX844628	[41]
Cercopinae	Cosmoscartini	*Cosmoscarta dorsimacula*	MG599490	[42]
**Cicadoidea**				
Cicadidae				
Cicadinae	Fidicinini	*Diceroprocta semicincta*	KM000131	Unpublished
Cicadettinae	Lamotialnini	*Magicicada tredecula*	MH937695	[43]
**Membracoidea**				
Aetalionidae				
Aetalioninae	Darthulini	*Darthula hardwickii*	KP316404	[44]
**Cicadellidae**				
Cicadellinae	Cicadellini	*Kolla paulula*	MW542170	Unpublished
	Proconiini	*Homalodisca vitripennis*	AY875213	Unpublished
Coelidiinae	Coelidiini	*Olidiana longisticka*	MN780582	[45]
		*Taharana fasciana*	KY886913	[46]
Deltocephalinae	Athysanini	*Abrus expansivus*	MK033020	[47]
		*Norvellina* sp.	KY039131	[48]
		*Tambocerus* sp.	KT827824	[49]
	Chiasmini	*Aconurella diplachnis*	OK105069	This study
		*Aconurella montana*	OK105070	This study
		*Aconurella prolixa*	MZ433366	[26]
		*Aconurella sibirica*	OK105071	This study
		*Doratura homophyla*	OK105076	This study
		*Doratura stylata*	OK105077	This study
		*Exitianus indicus*	KY039128	[25]
		*Exitianus nanus*	OK105078	This study
		*Gurawa minorcephala*	OK105072	This study
		*Leofa pulchella*	OK105073	This study
		*Nephotettix cincticeps*	KP749836	[25]
		*Nephotettix malayanus*	OK105079	This study
		*Nephotettix nigropictus*	OK105080	This study
		*Nephotettix parvus*	OK105074	This study
		*Nephotettix virescens*	OK105081	This study
		*Zahniserius cylindricus*	OK105075	This study
	Cicadulini	*Watanabella graminea*	MK234840	[50]
	Deltocephalini	*Alobaldia tobae*	KY039116	[48]
		*Maiestas dorsalis*	KX786285	[51]
	Drabescini	*Drabescoides nuchalis*	KR349344	[52]
		*Drabescus ineffectus*	MT527188	[53]
		*Roxasellana stellata*	MT527187	[53]
	Fieberiellini	*Fieberiella septentrionalis*	MW078430	[54]
	Macrostelini	*Macrosteles quadrilineatus*	KY645960	[55]
		*Macrosteles quadrimaculatus*	MG727894	[24]
	Mukariini	*Mukaria splendida*	MG813485	[56]
	Opsiini	*Hishimonoides recurvatis*	KY364883	Unpublished
		*Japananus hyalinus*	KY129954	[51]
		*Orosius orientalis*	KY039146	[48]
Deltocephalinae	Paralimnini	*Paralaevicephalus gracilipenis*	MK450366	[57]
		*Yanocephalus yanonis*	KY039113	[48]
	Penthimiini	*Penthimia melanocephala*	MT768010	[58]
		*Reticuluma hamata*	MN922303	[59]
	Scaphoideini	*Phlogotettix monozoneus*	MH427717	Unpublished
		*Phlogotettix* sp.	KY039135	[48]
		*Scaphoideus maai*	KY817243	[60]
		*Scaphoideus nigrivalveus*	KY817244	[60]
		*Scaphoideus varius*	KY817245	[60]
Eurymelinae	Idiocerini	*Idiocerus laurifoliae*	MH433622	[61]
		*Idioscopus clypealis*	MF784430	[62]
Evacanthinae	Evacanthini	*Evacanthus acuminatus*	MK948205	[63]
		*Evacanthus danmainus*	MN227166	[64]
Hylicinae	Hylicini	*Hylica paradoxa*	MW218660	[65]
	Sudrini	*Balala fujiana*	MW218661	[65]
Iassinae	Batracomorphini	*Batracomorphus lateprocessus*	MG813489	[66]
	Krisnini	*Krisna concava*	MN577635	[66]
Ledrinae	Ledrini	*Ledra auditura*	MK387845	[67]
		*Tituria sagittata*	MT610900	[68]
Megophthalminae	Agalliini	*Durgades nigropicta*	KY123686	[69]
		*Japanagallia spinosa*	KY123687	[69]
Mileewinae	Mileewini	*Mileewa alara*	MW533151	[70]
		*Ujna puerana*	MZ326688	[71]
Typhlocybinae	Erythroneurini	*Empoascanara dwalata*	MT350235	[72]
		*Empoascanara sipra*	MN604278	[73]
Membracidae				
Centrotinae	Centrotini	*Centrotus cornutus*	KX437728	[25]
	Leptobelini	*Leptobelus gazella*	JF801955	[74]
Smiliinae	Polyglyptini	*Entylia carinata*	KX495488	[75]

## Data Availability

The original contributions presented in the study are included in the article/Appendix A, further inquiries can be directed to the corresponding author/s.

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
