# Peer review of "Mitogenomic Analysis and Phylogenetic Implications for the Deltocephaline Tribe Chiasmini (Hemiptera: Cicadellidae: Deltocephalinae)"

_insects, 2024, doi:10.3390/insects15040253_

Round 1
Reviewer 1 Report
Comments and Suggestions for Authors
Author's list: do you really need to add the ORCID number in this section?
Review the verbal tenses along the manuscript, sometimes the present is used and then you change to past perfect or past perfect continuous.
Some parts of the text includded long sentences that difficults the understanding of the exposition.
Introduction:
Lines 75-78: maybe these are not the main characteristics of the mitochondrial genome in order to chose the molecule for phylogenetic analyses. I agree with the lack of recombination, but I miss the mention of the absence of histones that protect this genome, and then the relationship with the evolutionary rates or the matrilineal heritage. Also, it is important to pimpoint that no all the genome presents this high evolutionary rates, some genes or gene parts have it, others are highgly conserved across the phylogeny. When you mention the genome copy numbers, it is ok because we have many mitochondrions per cell and various copies of the mitochondrial chromosome per mitochondrion. However, this characteristic is important when performing analyses with critical samples and low quantities of DNA, because you increase the probability of detection and amplification.
Lines 85-87: Citations after the sentence with manuscript of the pest species cited.
Lines 87-90: the same than before, citations.
Please, highlight your objectives, not only mention the work you performed but also the aim of the study.
Review the sentences and the verbal tenses of the whole introduction.
Material and Methods.
Why do you preserve in EtOH and also stored at -80ºC? is not already preserved in ETOH?
Which guidelines did you use for species identification?
Did you check the DNA quality and integrity before the NGS assays (agarose geles or Nanodrop)? Which are the main requirements for this process (concentration, and quality ratios)?
Line 146: "COI gene was used as a reference genome" I don´t understand the sentence.
Results and discussion
Again, I feel that present and past are mixed along the text all the time and some sentences are really long.
Table 1 should be at the MyM section and not in the middle of resoults and discussion, also, can be part of the supplementary material because contains the Accession numbers of the organisms, not a real result to be considered.
Figure 1 and Figure 2: They are blurry and not really comented in the main text. Are they relevant in order to support this section? The same with figure 7.
Figures 8 and 9 are cited on page 15 and appear 4 pages later.
Why do you decide to remove the 3rd codon position in your analyses?
Which is your evolutionary model for the Bayesian analyses?
Why did you decided to concatenate your sequences for the study?
I miss the discussion in this section, you don´t explain and argue your results. There are just few "real" references in the text, the rest are related to the genomes employed in the analyses.
Author Response
Dear Reviewer,Thank you very much for your suggestions. According to your helpful comments and detailed suggestions, we have made a careful revision of the original manuscript. All changes made to the text are highlighted by using the track changes mode. The revised manuscript is uploaded. Below you will find our point-by-point responses to your comments and suggestions.
- Author's list: do you really need to add the ORCID number in this section?
Answer: We have included the ORCID of each author after their affiliation. Please note that these details may be subject to change during the production process by the production editor.
- Review the verbal tenses along the manuscript, sometimes the present is used and then you change to past perfect or past perfect continuous.
Answer: The manuscript has been revised for English proofreading and grammatical corrections.
- Some parts of the text includded long sentences that difficults the understanding of the exposition.
Answer: The manuscript has been revised to reduce the length of long sentences.
- Introduction
- Lines 75-78: maybe these are not the main characteristics of the mitochondrial genome in order to chose the molecule for phylogenetic analyses. I agree with the lack of recombination, but I miss the mention of the absence of histones that protect this genome, and then the relationship with the evolutionary rates or the matrilineal heritage. Also, it is important to pimpoint that no all the genome presents this high evolutionary rates, some genes or gene parts have it, others are highgly conserved across the phylogeny. When you mention the genome copy numbers, it is ok because we have many mitochondrions per cell and various copies of the mitochondrial chromosome per mitochondrion. However, this characteristic is important when performing analyses with critical samples and low quantities of DNA, because you increase the probability of detection and amplification.
Answer: Thank you for highlighting the key aspects of maternal inheritance and gene component conservation in the mtgenome. This section has been revised and rephrased accordingly. “The mitogenome is a suitable molecular marker for molecular evolution, phylogenetic relationships, and species identification due to its smaller size (approximately 14–20 kb), maternal inheritance, relatively high evolutionary rate, low levels of recombination, high genome copy numbers, and conservative gene components”.
- Lines 85-87: Citations after the sentence with manuscript of the pest species cited.
Answer: The references following each sentence have been updated accordingly.
- Lines 87-90: the same than before, citations.
Answer: The references following each sentence have been updated accordingly.
- Please, highlight your objectives, not only mention the work you performed but also the aim of the study.
Answer: Accepted. We have highlighted the aim of the study in the text as suggested by the reviewer.
- Review the sentences and the verbal tenses of the whole introduction.
Answer: Accepted. We have revised the text accordingly.
- Material and Methods
- Why do you preserve in EtOH and also stored at -80ºC? is not already preserved in ETOH?
Answer: The field samples were first preserved in ethanol and subsequently stored at -80ºC.
- Which guidelines did you use for species identification?
Answer: Prior to DNA extraction, the specimens were identified at species level based on external morphology and male genitalia from the following available source of literature [8–15].
- Did you check the DNA quality and integrity before the NGS assays (agarose geles or Nanodrop)? Which are the main requirements for this process (concentration, and quality ratios)?
Answer: Yes, we usually check the concentration and quality of DNA by Nanodrop and agarose.
- Line 146: "COI gene was used as a reference genome" I don´t understand the sentence.
Answer: This sentence has been deleted. As the raw data of new mitogenomes were assemble with the publish references of the tribe Chiasmini: Aconurella prolixa [MZ433366], Exitianus indicus [KY039128], and Nephotettix cincticeps [KP749836])
- Results and discussion
- Again, I feel that present and past are mixed along the text all the time and some sentences are really long.
Answer: We have reviewed the manuscript and have made the necessary corrections regarding the usage of present and past tenses, as well as addressing lengthy sentences.
- Table 1 should be at the MyM section and not in the middle of resoults and discussion, also, can be part of the supplementary material because contains the Accession numbers of the organisms, not a real result to be considered.
Answer: Table 1 has been moved to the Material and Methods section, and we would like to keep it in the main text file.
- Figure 1 and Figure 2: They are blurry and not really comented in the main text. Are they relevant in order to support this section? The same with figure 7.
Answer: The original pictures are of high quality. These figures serve as supporting visuals within the main text file.
- Figures 8 and 9 are cited on page 15 and appear 4 pages later.
Answer: The figure setting will be change before the formal publication, as in the current version there are several spaces after some figures and tables. We hope, the production editor will adjust all these issues in later stages.
- Why do you decide to remove the 3rd codon position in your analyses?
Answer: We used five different datasets (123PCG, 123PCG+2rRNA, 12PCG, 12PCG+2RNA, and AA) to evaluate their consistency with previous phylogenetic studies on this tribe. It is known that the third codon positions may suffer from mutation saturation, which can lead to noise in the phylogenetic analysis (Blouin et al., 1998; Breinholt & Kawahara, 2013). As expected in the present study, the datasets with third codon position removed [13 PCG sequences with the third codon position removed (P12), 13 PCG sequences with the third codon position removed, in addition to 2 rRNA sequences] yielded more consistent result with previous studies.
“In the PCG123-ML/BI and PCG123R-ML/BI trees, the topology is ((Zahniserius + Gurawa) + (Doratura + Aconurella)) + (Leofa + (Exitianus + Nephotettix)), while in the PCG12-ML/BI, PCG12R-ML/BI, and AA-ML/BI trees, the topology is ((Zahniserius + (Gurawa + (Doratura + Aconurella))) + (Leofa + (Exitianus + Nephotettix))). The latter topology has much higher support than the former and is more consistent with previous analyses based on combined mitochondrial and nuclear gene sequences [23] and anchored hybrid nuclear gene data [7], suggesting that the phylogenetic relationships within Chiasmini based on the mitochondrial P12, PCG12R, and AA datasets are more reliable.”
- Which is your evolutionary model for the Bayesian analyses?
Answer: We have provided the supplementary table of the partitioning schemes and best models. “PartitionFinder software 2.1.1 [36] was used to find the best partitioning scheme and and the best-fitting substitution models for the protein-coding genes' first, second, and third codon positions, and amino acid sequences (Table S1).”
- Why did you decided to concatenate your sequences for the study?
Answer: We concatenate the protein coding genes for phylogenetic analysis because it allows us to take advantage of more genetic information, which improves the detail and accuracy of the resulting phylogenetic trees. By integrating many genes into a single sequence, it provides more accurate evolutionary information as compared to individual gene trees.
- I miss the discussion in this section, you don´t explain and argue your results. There are just few "real" references in the text, the rest are related to the genomes employed in the analyses.
Answer: The results and discussions are presented under the same heading. We discuss our findings under each topic with the results.
With kind regards,
Yours sincerely
Yani Duan & Chris Dietrich
Reviewer 2 Report
Comments and Suggestions for Authors
In this study, the authors sequenced and annotated 13 complete mitogenomes of Chiasmini, which will contribute to phylogenetic studies within the tribe. However, there are notable areas in the analysis and English writing of the manuscript that require improvement. Specifically, if the newly sequenced mitochondrial genome does not exhibit any remarkable features, there is no need to extensively describe it. Instead, the focus of the article should be placed on phylogenetic analysis.
One such instance is found in line 32, where it is stated, "A comparative mitochondrial genome analysis." It is essential to clarify that these results stem from genome assembling and annotation rather than comparative mitochondrial genome analysis.
Additionally, the Summary lacks results from phylogenetic analysis, which are crucial and should be included.
Furthermore, content from lines 39-46 appears redundant in the abstract. The abstract should provide concise background information and highlight the main results.
To maintain consistency, it is advisable to consistently use either the abbreviation "mitochondrial genome" or its full term throughout the text.
Line 159 should include a citation for Mitos.
Regarding lines 185-196, clarification is needed on the models utilized in the phylogenetic analyses.
Comments on the Quality of English LanguageThere are notable areas in English writing of the manuscript that require improvement.
Author Response
Dear Reviewer,Thank you very much for your suggestions. According to your helpful comments and detailed suggestions, we have made a careful revision of the original manuscript. All changes made to the text are highlighted by using the track changes mode. The revised manuscript is uploaded. Below you will find our point-by-point responses to your comments and suggestions.
1. In this study, the authors sequenced and annotated 13 complete mitogenomes of Chiasmini, which will contribute to phylogenetic studies within the tribe. However, there are notable areas in the analysis and English writing of the manuscript that require improvement. Specifically, if the newly sequenced mitochondrial genome does not exhibit any remarkable features, there is no need to extensively describe it. Instead, the focus of the article should be placed on phylogenetic analysis.
Answer: We have revised the manuscript and improved the quality of the English. Considering that only 3 complete mitogenome sequences were previously available for the tribe Chiasmini, we think it is important to provide a more detailed comparative analysis even though the genomes of this tribe appear to be less variable than observed on some other leafhopper tribes. We provide additional discussion of the phylogenetic results as suggested.
2. One such instance is found in line 32, where it is stated, "A comparative mitochondrial genome analysis." It is essential to clarify that these results stem from genome assembling and annotation rather than comparative mitochondrial genome analysis.
Answer: This sentence has been modified as: The results shows that all 13 mitogenomes were composed of a circular, double-stranded molecule that consists of 37 genes with a total length ranging from 14,805 to 16,269 bp and a variable number of non-coding A+T-rich regions.
3. Additionally, the Summary lacks results from phylogenetic analysis, which are crucial and should be included.
Answer: The last paragraph of the summary now includes a brief summary of the phylogenetic analysis. “In the phylogenetic analysis, all subfamilies of the family Cicadellidae and tribes of the subfamily Deltocephalinae, were found to be monophylitic, except for the tribes Athysanini and Opsiini in Deltocephalinae were recovered as paraphylitic. The internal phylogenetic relationships within the tribe Chiasmini form a monophyletic group consisting of seven monophyletic genera arranged as follows: ((Zahniserius + (Gurawa + (Doratura + Aconurella))) + (Leofa + (Exitianus + Nephotettix)))”
4. Furthermore, content from lines 39-46 appears redundant in the abstract. The abstract should provide concise background information and highlight the main results.
Answer: The first sentence of the abstract has been deleted and the abstract has been modified accordingly.
5. To maintain consistency, it is advisable to consistently use either the abbreviation "mitochondrial genome" or its full term throughout the text.
Answer: The complete mitochondrial genome has been abbreviated as “mitogenomes” throughout the manuscript.
6. Line 159 should include a citation for Mitos.
Answer: Accepted. We have provided the citation.
7. Regarding lines 185-196, clarification is needed on the models utilized in the phylogenetic analyses.
Answer: Details regarding the models used in this study have been provided in the supplementary data.
Comments on the Quality of English Language
1. There are notable areas in English writing of the manuscript that require improvement.
Answer: Thank you for the recommendation. We have thoroughly checked and corrected the English.
With kind regards,
Yours sincerely
Yani Duan & Chris Dietrich
Reviewer 3 Report
Comments and Suggestions for Authors
The manuscript entitled “Mitogenomic analysis and phylogenetic implications for the deltocephaline tribe Chiasmini (Insecta: Cicadellidae: Deltocephalinae)” by Shah et al. presents an interesting contribution to systematics of Chiasmini. The manuscript is well written and fluent and I suggest you to have it published after some minor corrections that I report in the updated pdf of the manuscript.

Author Response
Dear Reviewer,Thank you very much for your suggestions. According to your helpful comments and detailed suggestions, we have made a careful revision of the original manuscript. All changes made to the text are highlighted by using the track changes mode. The revised manuscript is uploaded. Below you will find our point-by-point responses to your comments and suggestions.
The manuscript entitled “Mitogenomic analysis and phylogenetic implications for the deltocephaline tribe Chiasmini (Insecta: Cicadellidae: Deltocephalinae)” by Shah et al. presents an interesting contribution to systematics of Chiasmini. The manuscript is well written and fluent and I suggest you to have it published after some minor corrections that I report in the updated pdf of the manuscript.
Answer: Thank you for your acknowledgements and for considering our manuscript for possible publication in Insects. We have addressed all your comments in the revised manuscript.
- Maybe I will not start the introduction with this list of mitochondrial element, that I think is more appropriate for the MeM session.
Answer: We appreciate your suggestion. Since many molecular papers typically begin with a brief introduction about the mitogenome before introducing the main topic, we have followed a similar pattern in our manuscript. This approach does not affect the format of the present manuscript, so we prefer to retain the current version.
- I think is better to specify at the begginig of the sentence that you are tolking about Chine diversity, otherwise it is not clear
Answer: This sentence has been rephrased as “In China, Chiasmini comprises 39 described species in 11 genera:”
- I aslo add this part in ()
Answer: We have deleted these words from the text to avoid any confusion for the readers.
- As a reference genome or gene, you only mantion cox1.
Answer: This sentence has been deleted. As the raw data of new mitogenomes were assemble with the publish references of the tribe Chiasmini: Aconurella prolixa [MZ433366], Exitianus indicus [KY039128], and Nephotettix cincticeps [KP749836])
- I suggest the autors to explein better this sentence, is not so clear.
Answer: We have removed this sentence from the text to avoid confusion for the readers.
- I recommend replacing with morphological character or traits
Answer: Accepted. We have revised the text accordingly.
With kind regards,
Yours sincerely
Yani Duan & Chris Dietrich
Reviewer 4 Report
Comments and Suggestions for Authors
The article well written. The only thing I would like to be improved is adding information about the protocol used for NGS library preparation.
Author Response
Dear Reviewer,Thank you very much for your suggestions. According to your helpful comments and detailed suggestions, we have made a careful revision of the original manuscript. All changes made to the text are highlighted by using the track changes mode. The revised manuscript is uploaded. Below you will find our point-by-point responses to your comments and suggestions.
- The article well written. The only thing I would like to be improved is adding information about the protocol used for NGS library preparation.
Response: Thank you so much for your acknowledgements and accepting our manuscript. We have incorporated all your suggestions in the revised manuscript.
With kind regards,
Yours sincerely
Yani Duan & Chris Dietrich
Round 2
Reviewer 2 Report
Comments and Suggestions for Authors
The author has responded to the questions raised and the manuscript has seen significant improvement.
Comments on the Quality of English LanguageThere has been some improvement in English writing.